# A Meta-Analysis of the Effects of Comprehensive Sexuality Education Programs on Children and Adolescents

**DOI:** 10.3390/healthcare11182511

**Published:** 2023-09-11

**Authors:** Eun Joo Kim, Bitna Park, Seong Kwang Kim, Min Ju Park, Ju Yeong Lee, A Ra Jo, Min Jae Kim, Ha Neul Shin

**Affiliations:** Department of Nursing, Gangneung-Wonju National University, 150 Heungeop-myeon Namwon-ro, Wonju-si 26403, Gangwon-do, Republic of Korea; kimeju@gwnu.ac.kr (E.J.K.); ksk1677@naver.com (S.K.K.); 20218065@gwnu.ac.kr (M.J.P.); positive@gwnu.ac.kr (J.Y.L.); 20225216@gwnu.ac.kr (A.R.J.); 20225096@gwnu.ac.kr (M.J.K.); tlsgksmf215@naver.com (H.N.S.)

**Keywords:** Comprehensive Sexuality Education, meta-analysis, adolescent

## Abstract

Childhood and adolescence are crucial periods for developing one’s awareness of sexuality. Comprehensive Sexuality Education (CSE) during these stages is essential for overall growth, fostering healthy self-concepts, and addressing diverse sexual issues among children and adolescents globally. A meta-analysis was conducted to analyze the effectiveness of CSE programs. A literature search was performed on EMBASE, PubMed, CINAHL, Cochrane Library, and PsycInfo for studies published before 14 June 2023, and based on the Preferred Reporting Items for Systematic Reviews and Meta-Analyses guidelines. We used the Comprehensive Meta-Analysis (CMA, V4) software version 4.0 for the analysis and interpreted the effect sizes according to Cohen’s definition. Between 2011 and 2020, 21 studies on CSE were published, with the United States having the most publications (17). Of the 34 studies reviewed, 20 were randomized controlled trials. The primary population for CSE was middle/high school students (15), with the most frequent age range being 10–19 years (26). The overall effect size of CSE was significant (effect size = 1.31, *p* < 0.001), with cognition (effect size = 5.76, *p* < 0.001) being the most significant. CSE is an effective educational tool for children and adolescents with a significant impact on variables such as cognition and abstinence. It should be incremental from childhood and adolescence to adulthood.

## 1. Introduction

The awareness of one’s own sexuality is formed during childhood and adolescence and continues to develop and change throughout life [1]. Proper sexual education during childhood and adolescence is especially important for holistic growth and the development of healthy self-concepts [2]. Individuals establish independence and autonomy within the context of social and cultural environments at their adolescent stage, during which they are expected to develop major aspects such as the formation of an identity and the evolution of appropriate intimate sexual relationships with peers [3]. Although the body undergoes rapid maturation during adolescence, cognitive abilities may remain immature in terms of abstract thinking and social development [3]. This discrepancy can lead to unique problems such as increased vulnerability to sexually transmitted infections (STIs), as well as a potential for exhibiting abnormal behavior patterns during adolescence [4].

Sexual problems among children and adolescents vary in severity and prevalence worldwide [5]. In more industrialized countries like the United States, sexual initiation is increasingly occurring at early ages [6]. This phenomenon can be influenced by a myriad of factors, including socio-economic status, cultural and familial norms, and high social media usage, among others [7]. It is important to note that, while only 25% of adolescents aged 15–24 in the USA reported having had sexual intercourse, they account for half of the two million cases of STIs that occur annually, and unwanted pregnancies account for over half of teenage pregnancies, reaching as high as 82% [8]. In low- and middle-income countries, approximately 20% of adolescents have reported experiencing domestic sexual violence [9]. In developing countries, it has been reported that female adolescents in elementary and middle schools experience their first sexual encounter at an earlier age compared to their counterparts in more developed countries [10]. It is reported that 33% of women aged 12 to 25 have experienced violence, and 5% of them experienced sexual violence [11]. This issue of adolescent sexual problems is not confined to a single country. In 2009, the World Health Organization (WHO) began emphasizing the importance of reducing human immunodeficiency virus (HIV) among children and adolescents and preventing early pregnancy [12].

Comprehensive Sexuality Education (CSE) is based on a curriculum aimed at teaching the cognitive, emotional, physical, and social aspects of sexuality. It should be age-appropriate and culturally relevant [13]. The objective is to equip children and adolescents with knowledge, skills, attitudes, and values that can enhance their abilities, allowing them to make informed choices that consider their own and others’ well-being. It also aims to increase their understanding and protection of their own rights in life, as stipulated by the United Nations Convention on the Rights of the Child (UNCRC), and enable them to understand and protect their rights. This educational program was designed to help individuals understand sexuality and its impact on their lives more comprehensively. The United Nations Educational, Scientific and Cultural Organization (UNESCO) endorses CSE as a means of promoting healthy sexuality and reducing the negative consequences of risky behavior [13]. Overall, sex education entails teaching individuals to respect human dignity, protect themselves, value other people’s bodies, and show consideration for others, in addition to imparting knowledge related to physical and reproductive aspects [14].

Currently, CSE is being actively implemented worldwide, targeting children and adolescents, and positive outcomes are being reported. However, there are various limitations to effectively delivering CSE due to the comprehensive and extensive nature of its educational content, which should be age-appropriate, culturally relevant, and inclusive of unbiased information, skills, and attitudes, as outlined by UNESCO [13]. Moreover, the diverse limitations arise from the fact that the content of CSE is comprehensive and vast, making it challenging to provide the same education across different countries and cultures [15]. These challenges underscore the need for a thorough examination and evaluation of existing CSE programs, a task this study aims to undertake.

In this study, we aimed to conduct a meta-analysis of experimental design studies that provided CSE and examined its effects. The scope of this study encompassed CSE programs implemented worldwide, comprehensively analyzing the target population, educational methods, and content, as well as educational outcomes. Such an endeavor is expected to provide valuable insights for future endeavors in selecting target populations for CSE and choosing effective educational approaches.

## 2. Materials and Methods

### 2.1. Research Design

A meta-analysis was performed to examine and evaluate the intervention effects of a sex education program using CSE. This study followed the reporting guidelines of the Cochrane Handbook for Systematic Reviews of Interventions 5.1.0 and Preferred Reporting Items for Systematic Reviews and Meta-Analyses (PRISMA). It set key research questions, criteria for selecting and excluding the literature, data extraction processes, and search database scope and settings, assessing the quality of the literature, measuring the risk of bias, and conducting data and meta-analyses.

### 2.2. Selection and Exclusion Criteria for the Literature

To search for the literature, we set the following key questions: “What are the characteristics of studies that report the effects of CSE programs?” “What are the characteristics of participants in studies that verify the effects of CSE programs?” “What are the outcome variables of papers that verify the effects of CSE programs?” (In this context, outcome variables typically included measures related to sexual knowledge, attitudes, behaviors, and other related outcomes), and “What are the effects of CSE programs?” We based the inclusion criteria on the participants, intervention, comparison, outcomes, and study design. Specifically, we included studies that met the following criteria: (1) studies that intervened with sex education targeting children and adolescents, (2) studies that utilized CSE programs or included CSE programs as part of the intervention, and (3) studies that analyzed the effects of CSE programs. The exclusion criteria included the following: (1) studies that did not focus on sex education; (2) gray literature, such as poster presentations and abstracts; (3) studies that lacked analyzable data (i.e., studies that did not report statistics); (4) case reports; (5) qualitative studies; and (6) review studies.

### 2.3. Literature Search and Selection

The literature search was initially focused on all articles published up until 30 September 2022, using major databases including EMBASE, PubMed (MEDLINE), CINAHL, Cochrane Library, and PsycInfo. The search criteria targeted articles containing the terms “(Comprehensive Sexuality Education) AND (Effect OR RCT OR quasi OR experimental)” in their title or abstract. This initial search identified a total of 655 articles. After removing duplicates and articles unrelated to the study focus, 33 were selected for the meta-analysis.

Given the importance of including up-to-date research, especially considering that over six months had passed since our initial search cutoff in September 2022, we expanded our search parameters to encompass articles released from 1 October 2022 to 14 June 2023. This extended search yielded an additional 200 articles from the aforementioned databases. After a rigorous screening process and the removal of duplicates, one more article met our inclusion criteria. Consequently, the total number of articles selected for the meta-analysis amounted to 34, as detailed in Figure 1.

### 2.4. Data Evaluation

After the screening process, the final selection of studies included the authors, country, publication year, population, educational methods and content, funding status, and analyzed research outcomes. The research outcomes were categorized by grouping outcomes with the characteristics such as ‘attitude of condom use’ and ‘nontraditional gender attitudes’ were categorized under the ‘attitude’ outcome, while outcomes like ‘sexual health information’ and ‘STI knowledge’ were categorized under the ‘knowledge’ outcome category. Additionally, the content of CSE was categorized according to the classification method proposed in Miedema et al. study within their conceptual framework [15]. Miedema et al. identified the core components of CSE as follows [15]: (A) Rights, participation, and agency of young people: CSE aims to develop knowledge, attitudes, and life skills to protect the sexual and reproductive health and rights of adolescents. (B) Issues and practices related to sexual and reproductive health: CSE should address a wide range of issues and practices related to sexual and reproductive health. (C) Gender equality and power relations: this component emphasizes the impact of gender equality and women’s decision-making power. (D) Positive sexuality and respectful relationships: this component highlights the cultivation of healthy perceptions and attitudes toward sexual relationships among adolescents [15,16,17]. The researchers categorized and organized the studies into A, B, C, and D categories based on the educational purposes (educational content) of each study. Finally, two researchers with experience in conducting meta-analyses through the screening process and six additional researchers independently conducted quality evaluations based on a checklist. To improve the inter-rater reliability, two pre-training sessions were conducted using evaluation tools (Revised Cochrane Risk of Bias 2.0 tool). In cases of disagreement, a third party conducted additional quality evaluations, and a final consensus was reached through discussion. When evaluating the quality of the literature, randomized controlled trials were assessed using the Revised Cochrane Risk of Bias 2.0 tool from The Cochrane Collaboration, which evaluated randomization processes, departures from intended interventions, missing data on intervention outcomes, the measurement of intervention outcomes, and the selective reporting of study results. Nonrandomized comparative studies were evaluated using the Risk of Bias in Nonrandomized Studies of Interventions (ROBINS-I) tool, which assessed eight items: participant selection, the comparability of groups, confounding variables, exposure measurement, outcome evaluation, the blinding of assessors, the incomplete reporting of data, and the selective reporting of results. The NECA systematic literature review manual was consulted in the usage of both RoB 2.0 and ROBINS-I [18].

### 2.5. Data Analysis

All statistical analyses related to the effect sizes of the comprehensive gender and sexuality education program variables were conducted using the Comprehensive Meta-Analysis (CMA, V4) software version 4.0 [19].

We interpreted the effect sizes based on Cohen’s d definition [20], where values of 0.2–0.5 are considered small, 0.5–0.8 as medium, and ≥0.8 as large, with a significance level set at 95%. Given that meta-analyses involve aggregating results from various studies to interpret the effect size, assessing the homogeneity or heterogeneity of the research becomes crucial. To determine this, we referred to the I^2^ values. An I^2^ value range of 0–24% suggests no heterogeneity, 25–49% indicates low heterogeneity, 50–74% represents moderate heterogeneity, and values ≥75% signify high heterogeneity [21]. However, Borenstein has cautioned against a blanket application of these traditional interpretations, emphasizing the importance of context-specific evaluations [22,23]. They advised using the fixed-effect model when assuming that there is the same effect size across all studies, suitable for low heterogeneity [24]. On the other hand, when presuming that different studies might have varying effect sizes, he recommended analyzing based on the random-effects model [24]. Accordingly, while we employed the random-effects model for analyses indicating heterogeneity of 75% or higher, in accordance with Borenstein’s guidance [22,23], we also incorporated the prediction interval to provide a more detailed understanding of both the homogeneity/heterogeneity and the magnitude of the effect size.

To probe for publication bias, both funnel plot analysis and the trim-and-fill method were utilized [25]. Publication bias was suspected if the effect size, specifically referring to the adjusted measure that quantifies the magnitude of the intervention’s impact after considering potential publication bias using the trim-and-fill method, deviated by 10% or more compared to the prior estimate [25]. This adjusted effect size provides a more accurate representation of the true effect when accounting for any potentially missing studies. This method involves removing asymmetric effect sizes from the original funnel plot (trimming) and then estimating missing studies presumed to be left out, relying on studies that fill in the gaps, to achieve symmetry around the newly calculated mean effect size [26].

## 3. Results

### 3.1. General Characteristics of the Selected Studies

This study analyzed 34 articles (Studies List S1); the general characteristics of the articles are presented in Table 1. The publication years of the articles ranged from before 2000 to 2023, with most of them being published between 2011 and 2020 (*n =* 21, 61.8%). Most of the articles—17 (50.0%)—originated from the USA, followed by China with 5 (14.7%). The study designs comprised 20 (58.8%) randomized controlled trials and 10 (29.4%) quasi-experimental studies. The study duration had the highest occurrence of 11 (32.4%) cases within the range of one to two years. Additionally, 23 (67.6%) studies received funding, and component A of CSE was covered in 13 (38.2%) articles, making it the most common focal point, followed by component B, which was covered in 9 (26.5%) articles. Finally, the quality for 21 articles (low 85.7%, some concerns 14.3%) was assessed using RoB and RoBINS-I for 13 articles (low 92.3%, some concerns 7.7%).

### 3.2. Characteristics of the Educational Target of Comprehensive Sexuality Education

Table 2 presents the characteristics of educational targets. In terms of population, most were middle and high school students with 15 articles (44.1%), followed by adolescents with 6 articles (17.6%). The age range of 10–19 years was the most prevalent with 26 articles (76.5%). Regarding race/ethnicity, Hispanic/Latino and White were each prevalent in 11 articles (12.2%), followed by African American with 9 articles (10.0%). The gender category of female/male was the most prevalent in 24 articles (70.6%), while male had the lowest representation with 2 articles (5.9%). The number of enrolled participants and the number of randomized participants were both less than 1000 in 14 articles (41.2%) each, making it the highest proportion for each category. Additionally, the number of analyzed participants was less than 1000 in 18 articles (52.9%), which was the highest percentage.

### 3.3. Characteristics of Outcome Variables in CSE

The categorization of the CSE outcome variables revealed that knowledge (*n =* 33, 14.1%), unprotected sexual activity (*n =* 24, 13.7%), and attitude (*n =* 24, 10.1%) were the most common, whereas body image, capacity, self esteem, and STD testing (*n =* 1, 0.4%) were the least common (Table 3).

### 3.4. Effects of Comprehensive Sexuality Education

In our study, we examined heterogeneity, which was assessed using the *Q*-value and the I² statistic in a random-effects model. The *Q*-value tests the null hypothesis that all studies included in the analysis share a common effect size. This value was 2809.324 with 33 degrees of freedom, and the *p*-value was less than 0.001. Thus, we confirmed that the true effect sizes vary among all the studies included in the meta-analysis. Furthermore, the I^2^ statistic was 99%, indicating that approximately 99% of the variance in the observed effects can be attributed to true effects rather than sampling error. This high I^2^ value suggests a substantial level of heterogeneity among the observed effects. However, as mentioned in the two papers we previously reviewed [22,23], an I^2^ value of 99% does not necessarily imply a high level of heterogeneity. Therefore, it is difficult to fully understand the actual degree of heterogeneity using only the I^2^ value. In addition, our study estimated the prediction interval to be between 0.55 and 3.13, assuming that the true effects follow a normal distribution. This interval can be useful for predicting the range of effect sizes that will be observed in future studies. The wider the prediction interval, the greater the difference in true effect sizes between studies. This provides a range of results that considers heterogeneity, which can help compensate for the limitations of the I² value in measuring heterogeneity. Therefore, a high level of heterogeneity was observed in this study, indicating that the true effect sizes vary among the various studies [27,28,29,30,31,32,33,34,35].

The effect size of the study by Castellanos [36] (OR 3.57; CI 2.09–6.09), followed by the study by Hong [37] (OR 3.42; CI 2.54–3.61) had the largest effect sizes. On the other hand, studies such as Booth’s [17] (OR 1.00; CI 0.88–1.14), Degue’s [38] (OR 0.72; CI 0.43–1.22), and Hegdahl’s [39] (OR 1.03; CI 0.97–1.1) showed no significant effects. Additionally, the studies by Grossman [40] (OR 0.38; CI 0.29–0.49) and Dulli [41] (OR 0.78; CI 0.66–0.91) demonstrated a statistically significant negative effect. We found that CSE had a significant effect overall (OR 1.31; CI 1.13–1.51) (Figure 2).

Among the 25 outcome categories, cognition showed the largest effect size, as measured by the odds ratio (OR 5.76; CI 3.67–9.06). Additionally, the categories of abstinence (OR 2.90; CI 1.25–6.72), knowledge (OR 2.00; CI 1.49–2.67), self-efficacy (OR 1.76; CI 1.08–2.87), attitude (OR 1.76; CI 1.38–2.25), and intention (OR 1.29; CI 1.00–1.66) were positively enhanced. It was effective in delaying the onset of sexual behavior (OR 0.37; CI 0.16–1.86). Furthermore, pregnancy, assessed for its occurrence, showed a low odds ratio (OR 0.06; CI 0.03–0.14), indicating its significance in preventing pregnancy (Table 3) (Figure 3).

We conducted a meta-analysis to investigate the effects of different components of CSE. Our analysis revealed significant effects for components A (OR 1.81; CI 1.54–2.12) and D (OR 1.29; CI 1.01–1.65). However, it is important to note that components B (OR 1.05; CI 0.77–1.44), C (OR 1.15; CI 0.84–1.57), and others did not show significant effects. Furthermore, we observed high heterogeneity in the results for components A (*Q* = 3561.28, *p* < *0*.001, I^2^ =98.32) and D (*Q* = 333.78, *p* < *0*.001, I^2^ =92.81), indicating substantial variability among the included studies (Figure 4).

### 3.5. Publication Bias in the Meta-Analysis of Comprehensive Sexuality Education

We visually assessed the degree of symmetry of the 34 selected studies using a funnel plot to determine whether there was any publication bias. No publication bias was found because the plot was generally symmetrical with respect to the middle line. The result of the trim-and-fill analysis shows that there were 0 studies trimmed, indicating that there were no papers considered biased and in need of modification or removal, indicating that there were no changes in the effect size and the confidence intervals remained unchanged as well. Therefore, it can be inferred that the publication bias of the included studies had little impact and the original results were maintained in this study (Figure 5).

## 4. Discussion

CSE is an educational program aimed at providing knowledge, skills, attitudes, and values that enable children and adolescents to recognize and respect their health, well-being, and dignity, as well as form social and sexual relationships. It also aims to help individuals understand and protect their rights [13]. Based on this, we conducted a meta-analysis of 53 studies on CSE’s effectiveness to examine its content and effects.

The review results showed that the publication of CSE-related papers has increased since the 2010s. Despite ongoing issues related to child and adolescent sexual problems, the development of CSE in 2009 and the revised guidelines published in 2018 [13] have led to a significant increase in related research. This trend is most apparent in the United States; about 50% of young people are infected with STIs every year, with approximately 20% newly diagnosed with HIV [68], highlighting the need for effective sex education. In 2010, the Obama administration announced a new national strategy for teenage pregnancy prevention that focused on CSE, resulting in an increase in research funding for sex education programs and their impact [69]. It should be noted that the political stance on sex education, particularly those advocating for abstinence-based education, could significantly influence these statistics. Such policies may impact the effectiveness of CSE and subsequently affect the rates of teenage pregnancy and sexually transmitted infections. Nevertheless, the severity of child and adolescent sexual problems in low-income and developing countries necessitates active support and attention from the international community.

Most of the interventions implemented in the analyzed studies had durations of less than a year or between one and two years. Conversely, the Center for Disease Control and Prevention (CDC) recommends providing CSE programs suitable for the development and comprehension of key concepts from late elementary or early middle school to high school for a minimum of three years [70]. Therefore, to effectively enhance sex education, it is crucial to conduct studies on CSE that span over three years or longer.

Middle and high school students were the most represented in the population section. CSE is typically provided to middle and high school students who are exploring their sexuality and can acquire knowledge about sexual health and relationships during this time. This can reduce negative outcomes such as unplanned pregnancies and sexually transmitted infections [13]. Additionally, most middle and high school students are required to attend school; schools can provide CSE to many students, which may explain why numerous studies have targeted this age group.

The study by Castellanos [36], which had a high effect size, was conducted in the United States and included high school students. The study focused on improving the knowledge and acceptance of the HPV vaccine, with peer-to-peer education lasting 30 min. The results showed a significant improvement in knowledge of the HPV vaccine after the training; however, publication bias was a concern in this study. Another study with a large effect size by Hong [37] followed elementary school students for six years. The study examined a range of variables including family and marriage, life skills, rights, self-protection, genitalia, hygiene and health, and adolescent development. The large effect size is likely due to the long duration of the study and the comprehensive examination of the effects on a wide range of variables.

The outcome variables in CSE were predominantly related to knowledge. This seems appropriate as the goal of CSE is to effect changes in knowledge and attitudes, providing a deeper understanding of one’s sexuality [13]. Specifically, the enhancement of sexual knowledge impacts individual behavior and helps one better understand their rights and responsibilities when making sexual decisions [71,72]. It is known that the changes in knowledge brought about through CSE have a direct positive effect on the responsibility for children and adolescents’ sexual behavior [13], thus making it one of the most crucial effects of education. In the current context, where ambiguous information related to sex is easily accessible [73,74], comprehensive sexuality education, which equips children and adolescents with the knowledge and skills for responsible decision making, not just the mere dissemination of sexual knowledge, can be considered essential for effective health promotion [42,75,76].

This study confirmed that CSE is a highly effective educational method for children and adolescents. Furthermore, cognition exhibited the largest effect size. This reaffirms the crucial role of CSE in improving awareness of sexuality. However, there are significant differences between countries and cultures in terms of the necessity and content of sex education [13]. Due to these differences, children and adolescents are likely to be exposed to inconsistent sex education. The most effective study included in our analysis reported that the perception and attitude towards the sexuality of immigrant children who received CSE for six years improved [37]. In other words, it was described that applying CSE consistently and long-term can be beneficial for their current and future health and well-being. Additionally, variables that delay sexual onset and pregnancy showed a significant effect size. These results align with previous studies that suggest delaying sexual intercourse and thus preventing unsafe sex is effective in reducing additional issues such as teen pregnancy, STDs, and HIV infection [70]. To maintain these effects, it is important for CSE to be provided consistently and long-term [13]; rather than efforts by individuals or certain organizations, there is a need for expansion at the state and national levels [77].

Finally, the contents of the CSE in the analyzed studies was categorized into four domains according to Miedema et al. classification, and their effectiveness was validated [16]. Most studies focused on education related to category A, which aims to educate children and adolescents about their sexual rights, including the right to sexuality, sex-related health, and the responsibility to respect the rights of others. The pooled effect size was significantly effective. The educational content in category A mainly aims to promote knowledge-, attitude-, and perception-related changes, and is considered to be a fundamental part of CSE [13]. Considering the characteristics of children and adolescents at different developmental stages, education to establish such values correctly should be an essential component of CSE. The second significant effect was observed in category D, which was related to education on positive sexuality and respectful relationships. In other words, it aims to illustrate that positive sexuality is not merely taboo sexual pleasure but entails acts of respect, consideration, and ultimately, equal and reciprocal values [13]. This study constituted information on various beliefs and social stigmas related to sexuality, as well as the need for attention and consideration regarding vulnerability related to race, disabilities, and sexual orientation. This study aimed to confirm changes in knowledge and attitudes through such education. However, CSE did not comprehensively cover everything in the category. Instead, it tended to deliver only fragmented training content. A more systematic and comprehensive training program should be developed, not just for the training categories that were found to be effective.

Regarding the importance of CSE, UNESCO emphasizes comprehensive education that allows children and adolescents to have healthy and safe social and sexual relationships, starting with respect for oneself and others [13]. According to this study, the effectiveness of CSE was statistically verified, particularly in terms of cognition, knowledge, intention, and self-efficacy, and it affected the behavior related to pregnancy and delayed sexual intercourse.

These results highlight that CSE, as supported by the review and meta-analysis, can contribute to improving the quality of life and sexual health of adolescents by equipping them with knowledge and skills related to healthy sexual behavior and relationships. It is important to note that the findings mentioned primarily pertain to the studies reviewed. This study serves to raise awareness among policymakers, educators, parents, and adolescent health experts about the significance and effectiveness of CSE.

In this study, we evaluated heterogeneity among studies on the effects of Comprehensive Sexuality Education (CSE) using the Q-value, I² statistic, and prediction interval [32]. Our findings reveal extensive heterogeneity, suggesting that CSE’s effects can significantly vary depending on multiple factors, including education content and methods, as well as cultural and social backgrounds [13]. Moreover, the prediction interval, ranging from 0.55 to 3.13, suggests the potential for variation in the effect size of CSE in future studies, reflecting the high heterogeneity we observed. These findings underscore the importance of caution when evaluating the effects and interpreting the results of future CSE studies.

The limitations of this study are as follows. First, despite our efforts to integrate various studies, the selected studies may not represent all CSE programs. Second, the high heterogeneity observed in the components indicates that the results should be interpreted carefully, and additional research is required to understand the factors contributing to this heterogeneity.

Future research should identify the causes of the heterogeneity observed in the meta-analysis, explore the causes of variation to ensure the effective implementation of CSE, and elucidate the specific effects of each component of CSE to effectively implement it for children and adolescents. Based on the results of this meta-analysis, it is crucial to identify the overall effectiveness and importance of CSE and focus on developing and implementing CSE that emphasizes the significant effects of each component, leading to positive outcomes on the health and well-being of children and adolescents.

## 5. Conclusions

CSE is an important educational tool that aims to provide knowledge, skills, attitudes, and values, enabling individuals to recognize and respect social and sexual relationships and understand and protect the rights, health, welfare, and dignity of children and adolescents. This study showed that CSE positively affects various areas such as sexual health, knowledge, and attitudes of children and adolescents. However, the effectiveness of CSE may vary depending on the operating method, educational content, and target students, and appropriate programs should be developed considering these differences in individual educational settings. Moreover, careful reviews and modifications of educational content and methods are necessary to maximize the effectiveness of CSE. Efforts to increase the effectiveness of CSE through systematic evaluations and improvements are necessary in educational settings. The results of this study are expected to provide additional insights on the importance and effectiveness of CSE to policymakers, educators, parents, and adolescent health professionals. Additionally, the study is expected to serve as a basis for the systematic development and implementation of CSE.

## Figures and Tables

**Figure 1 healthcare-11-02511-f001:**
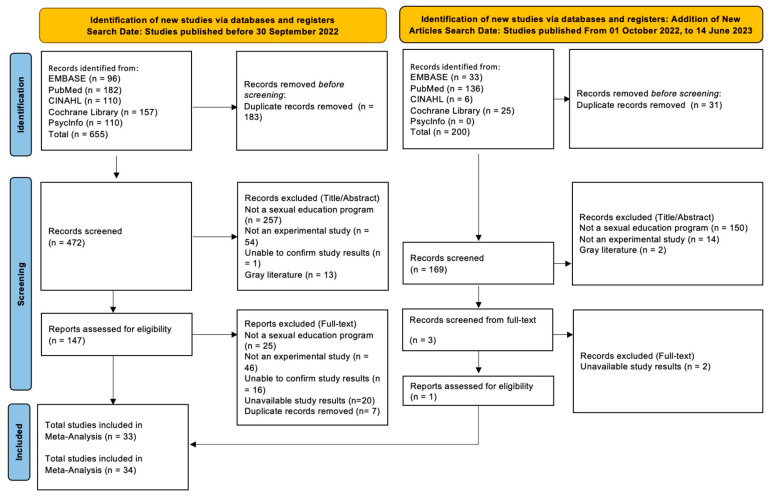
PRISMA 2020 flow diagram for the meta-analysis of this study.

**Figure 2 healthcare-11-02511-f002:**
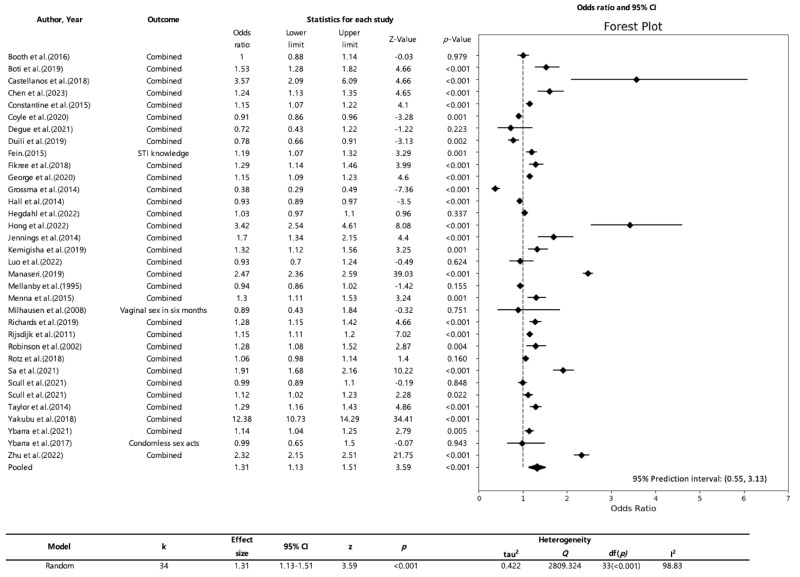
Total odds ratio of Comprehensive Sexuality Education (*n =* 34) [17,36,37,38,39,40,41,42,43,44,45,46,47,48,49,50,51,52,53,54,55,56,57,58,59,60,61,62,63,64,65,66,67].

**Figure 3 healthcare-11-02511-f003:**
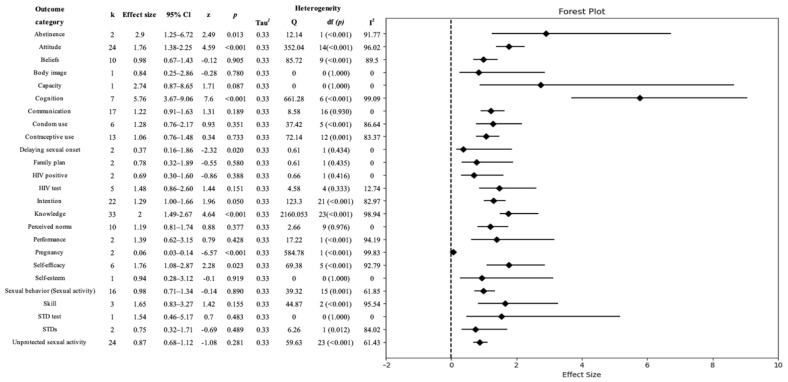
Effect size of Comprehensive Sexuality Education variables (*n =* 34). Note: HIV, human immunodeficiency virus; STD, sexually transmitted disease.

**Figure 4 healthcare-11-02511-f004:**
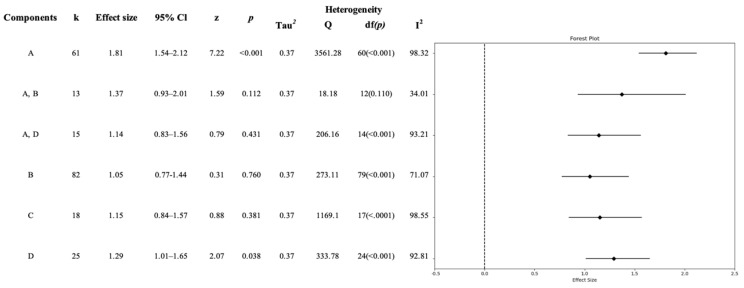
Effect size of Comprehensive Sexuality Education components (*n =* 34). Note: A. Young people’s rights, participation, and agency. B. Sexual- and reproductive-health-related concerns and practices. C. Gender equality and power relations. D. Positive sexuality and respectful relationships.

**Figure 5 healthcare-11-02511-f005:**
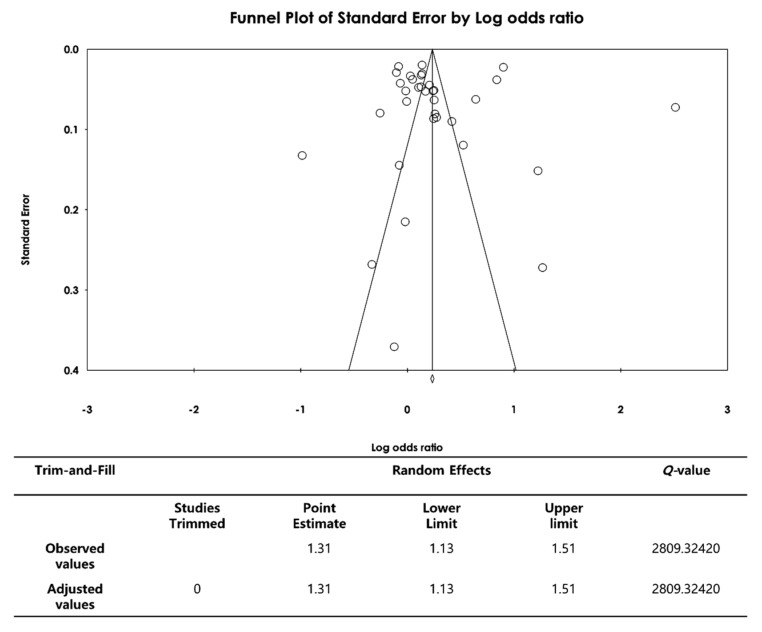
Funnel plot and trim-and-fill for the publication bias of the studies.

**Table 1 healthcare-11-02511-t001:** General characteristics of the selected studies (*n =* 34).

Characteristics	Categories	Subcategories	k	Percentage
Year of publication	Before 2000	-	1	2.9
	From 2001 to 2010	-	2	5.9
	From 2011 to 2020	-	21	61.8
	From 2021 to 2023	-	10	29.4
Country of Publication	North America (*n =* 17)	USA	17	50.0
	South America (*n =* 1)	Dominican Republic	1	2.9
	Europe (*n =* 2)	UK	1	2.9
		Norway (Zambia)	1	2.9
	Asia (*n =* 5)	China	5	14.7
	Oceania (*n =* 3)	Ethiopia	3	8.8
	Africa (*n =* 6)	Ghana	1	2.9
		Kenya	1	2.9
		South Africa	2	5.9
		Uganda	2	5.9
Study design	Experimental/Cohort study (*n =* 47)	RCT/experimental study	20	58.8
		Quasi-experimental study	10	29.4
	Others (*n =* 6)	Mixed methods	2	5.9
		Non-categories	2	5.9
Study duration	<1 year	<6 months	4	11.8
		6–11 months	3	8.8
	1–2 years	12–17 months	9	26.5
		18–24 months	2	5.9
	2–3 years	25–30 months	1	2.9
		31–36 months	1	2.9
	3–4 years	48 months	5	14.7
	>5 years	60 months	1	2.9
		72 months	1	2.9
	Not described	-	7	20.6
Funding	Yes	-	23	67.6
	No	-	11	32.4
Component of CSE	A	-	13	38.2
	B	-	9	26.5
	C	-	4	11.8
	D	-	2	5.9
	A, B	-	2	5.9
	A, D	-	3	8.8
	B, D	-	1	2.9
Quality evaluation	RoB	Low	18	85.7
		Some concern	3	14.3
		High	0	0
	RoBINS-I	Low	12	92.3
		Some concern	1	7.7
		High	0	0

Note: A. Young people’s rights, participation, and agency. B. Sexual- and reproductive-health-related concerns and practices. C. Gender equality and power relations. CSE, Comprehensive Sexuality Education. D. Positive sexualities and respectful relationships. RoB, Risk of Bias for randomized trials. RoBINS-I, Risk Of Bias in Non-Randomized Studies of Interventions. RCT, randomized controlled trials.

**Table 2 healthcare-11-02511-t002:** Characteristics of the educational target of Comprehensive Sexuality Education (*n =* 34).

Characteristics	Categories	Subcategories	k	Percentage
Population	Student	Middle and high school students	15	44.1
		Elementary school students	1	2.9
		College students	1	2.9
		11–25-year-old school students	1	2.9
	Adolescents	-	6	17.6
	Sex	Male	1	2.9
		Female	4	11.8
		Cisgender female Sexual minority	1	2.9
	Patients	-	1	2.9
	Female sex workers	-	3	8.8
Age	10–19 years	-	26	76.5
	13–25 years	-	1	2.9
	32 years		1	2.9
	Not described	-	6	17.6
Race/ethnicity	Asian	Asian	5	5.6
		Chinese	1	1.1
		Japanese	1	1.1
	American	Asian-American	1	1.1
		American Indian	5	5.6
		Chicano	1	1.1
		Hispanic/Latino	11	12.2
		African American	9	10
		Hawaiian	3	3.3
		Alaskan Native	1	1.1
		Alaska Native Multiracial	1	1.1
	African	Amhara	2	2.2
		Oromo	1	1.1
		Ashantis	1	1.1
		Kenyan	1	1.1
		South African Asians	1	1.1
		Ugandan	2	2.2
		Zambian	1	1.1
	European	Ukrainian	1	1.1
	Others	White	11	12.2
		Black	5	5.6
		Others	6	6.7
		Other Pacific Island	3	3.3
		Biracial	1	1.1
		Spanish descent	1	1.1
		Not described	14	15.6
Sex/gender	Female	-	8	23.5
	Male	-	2	5.9
	Female/male	-	24	70.6
Number of enrolled	<1000	-	14	41.2
	1000–3000	-	6	17.6
	>3000	-	6	17.6
	Not described	-	8	23.5
Number of randomized	<1000	-	14	41.2
	1000–3000	-	6	17.6
	>3000	-	6	17.6
	Not described	-	8	23.5
Number of analyzed	<1000	-	18	52.9
	1000–3000	-	8	23.5
	>3000	-	3	8.8
	Not described	-	5	14.7

**Table 3 healthcare-11-02511-t003:** Characteristics of outcome variables in Comprehensive Sexuality Education (*n =* 34).

Categories	Outcome Variables	k	Percentage
Abstinence	Abstinence from sex/Abstaining from penile–vaginal sex	2	0.9
Attitude	Attitudes towards using condoms, healthy sexual behavior, homosexuality, gender roles, and CSE	24	10.1
Beliefs	Beliefs about causes of pregnancy, HIV, STIs/Acceptance of gender roles, dating violence, and rape myths	10	4.8
Body image	Body image	1	0.4
Capacity	CSE capacity	1	0.4
Cognition	Risk perception towards HIV, STIs, pregnancy/Perceived susceptibility, severity, barriers, benefits	7	3.1
Communication	Communication with parents, partner, boyfriend, or girlfriend regarding sexual health	17	8.4
Condom use	Inconsistent or consistent condom use	6	3.1
Contraceptive use	Contraceptive use/Currently using birth control	13	5.3
Delaying sexual onset	Delaying vaginal sex (boys and girls)	2	0.9
Family plan	Family planning referrals (male and female)/Married/Cohabiting	3	1.3
HIV positive	HIV-positive/HIV-negative	2	0.9
HIV test	HIV tested/Not HIV tested/Willingness to HCT	5	2.2
Intention	Intention to use a condom, contraception, protection, prevent pregnancy/Intent for bystander intervention	22	9.7
Knowledge	Knowledge of contraceptive methods, sexual, STD testing, causes of STIs, non-causes of HIV, pregnancy, STI prevention	33	14.1
Perceived norms	Perceived community norms regarding contraceptives, condom use, gender equitability, teen sex description, dating violence, gender roles, contraception, protection norms	10	4.4
Performance	Performance behavior to escape and avoid situations where unwanted sex could happen	2	0.9
Pregnancy	Pregnancy since program enrollment/Was or is pregnant/Never been pregnant	2	0.9
Self-efficacy	Self-efficacy towards condom use/Self-efficacy in dealing with coercive sex/Contraception and protection efficacy	6	3.1
Self esteem	Self esteem	1	0.4
Sexual behavior (sexual activity)	Ever had sex/Had vaginal sex/Rights with steady partner/Number of condomless sex acts/Limiting sexual partner only to one	16	8.4
Skill	CSE skill	3	1.3
STD test	Receiving more than one joint testing	1	0.4
STDs	Reported STD symptoms	2	0.9
Unprotected sexual activity	Unprotected sexual activity/More than one sexual partner/Engage in transactional sex/Sexual violence perpetration or victimization/Teen sexual intentions/Unmet contraceptive needs	24	13.7

Note: CSE, Comprehensive Sexuality Education; HCT, human immunodeficiency virus counseling and testing; HIV, human immunodeficiency virus; STD, sexually transmitted disease; STI, sexually transmitted infection.

## Data Availability

The data used to support the findings of this study are under ethical restrictions and cannot be made publicly available. Data are available from the corresponding author for researchers who meet the criteria for access to confidential data.

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
