# Peer review of "A Meta-Analysis of the Effects of Comprehensive Sexuality Education Programs on Children and Adolescents"

_healthcare, 2023, doi:10.3390/healthcare11182511_

Round 1

Reviewer 1 Report (Previous Reviewer 1)

Dear Authors,

The new version of your manuscript is in much better shape. I still found a few things you may work on to improve even further your manuscript. These are detailed as follow:

ABSTRACT
I insist, the goal of the study is no "background". But, I see the authors have made their choice.

INTRODUCTION

Line 39 = I believe many other factors are at play in sexual debut than just sexual problems, educations, or lack thereof.

Line 46 = earlier than?

Line 63 = I suggest to begin the sentence with “overall, …”

Line 66 = whereas I agree, using “should” is pitching. Sex Ed could start later and catch up. I believe there are many people who never get Sex Ed and live good/rewarding sexual lives. The word needs to be replaced.

Line 75 = I believe there is a sentence missing that would wrap up the problematization the authors are trying to make to justify their study. For instance, I would assume that a good piece of evidence able to ascertain the impact of the already conducted sex ed programs is a review. Therefore, in this study…. Something along the lines, something that establishes more explicitly the problematization the authors are making in the paragraph between the lines 76-81.

METHODS

The information between Line 110 from “including 96 from EMBASE…” until the period in Line 115 is unnecessary. It is already contained in Figure 1. Similarly, much of the info in that section can be taken out to avoid repetition. The Figure explains it, use it. Use the text to summarise.

Line 153 = Which evaluation tools?

Line 163 = the NECA system manual should be referenced

Line 171-173 = there should be at least one comma in this long sentence.

Line 178 = This sentence is redundant with the information given in the previous sentence. Unknowledgeable readers on I2 still do not know whether heterogeneous studies or homogenous is what is expected, ideal, best, etc.

Line 184 = which effect size? How to interpret the “effect size”?

RESULTS

Line 202 = there are two final periods in that sentence.

Table 2 = There is a comma after “cisgender female”, while “sexual minority” does not have a capital letter in “sexual” (following the author’s own format).

TABLES IN GENERAL = I would suggest that the furthest most left column labels is at the top of the cell. Being able to identify which categories belong to it when the furthest most left label is centered makes it unnecessarily harder.

Figure 2 is unacceptably blurry. Please, do correct it! Also, the POOLED statistics and diamond should be bigger, noticeable.

Figure 3 is also blurry, less than Figure 2, but still blurry. So is Figure 4.

If the authors magnify Figure 5, they will see why the rest are blurry in comparison to this one.

Sincerely,

None. As always, I encourage the authors to have an experience, hopefully English native writer, to read it before sending the last version of your manuscript. Your last version was in good shape, though.

Author Response

Reviewer 2 Report (New Reviewer)

Very interesting review.

Follow methodological guidelines properly.

Well described results.

Excellent discussion.

Conclusion very well.

Congrats.

Author Response

Reviewer 3 Report (New Reviewer)

In abstract define CSE the first time you use the acronym

In the abstract indicate which effect size you are using

In the introduction, first paragraph last sentence is unclear.

Is there a better term than “advanced” for countries like the US? (Not sure what the advanced refers to – perhaps something like “more industrialized” would be better?)

The third paragraph of the introduction contains only one sentence so perhaps this sentence could be placed in the paragraph above?

In the fourth paragraph of the introduction the first sentence has some grammatical issues.

Material from the fourth and fifth paragraphs of the introduction should be combined and integrated since it all relates to defining CSE.

In this sentence: “In this study, we aimed to conduct a meta-analysis of experimental design studies 76 that provided CSE and examined their effects.”  Should it be its effects instead of their?

You should define the time period used for the literature search.

You should more clearly define the important outcome variables in both the method section and the results – there should be more theory provided to support the relationships studied

My understanding was that EITHER a fixed or random effects model should be chosen to analyze the data. You should provide a reference to support your choice and/or provide more validity information.

No comments

Round 2

Reviewer 3 Report (New Reviewer)

I was an original reviewer. I still think the data are not adequately described. Are effect sizes d values or odds-ratios? The outcome variables need to be clearly described, and then studies compiled for each of the outcome variables clearly laid out in the tables (the current tables are confusing).

Author Response

This manuscript is a resubmission of an earlier submission. The following is a list of the peer review reports and author responses from that submission.

Round 1

Reviewer 1 Report

Kim et al., provided a review and meta-analysis on comprehensive sexual education programs on children and adolescents.  Their study is well presented, designed, and executed. I have minor details I believe the authors will not have problem resolving and addressing that, I hope, lead to improve their manuscript. They are as follow:

Title

                I believe “sexual” education is better suited.

Abstract

Line 12: A study goal is not its rationale’s background. If background is required, then it should be along the lines of the context of the study.

Introduction

Line 32: it may not hurt to add that sex education should be age-appropriate and culturally relevant.

Line 42: “the awareness of one’s own sexuality…” is better suited than “sexual consciousness”. Such sentence should have a reference. I recommend Pfaus et al., 2012 Figure 9 to see how this process is developed. 10.1007/s10508-012-9935-5

Line 58: the authors used in line 56 the term, they should use the acronym STI there.

Line 61: I think, given the topic of the study, that the authors should disclose the age of the women whom they speak in the citation.

Line 80-82: I believe the study has much more value than thosegood reasonsmentioned here, especially those regarding the meta-analysis.

Line 87-97: To identify…

Line 98: whereas what is described in this section is helpful and valuable, I think it should be embedded in the previous section, for it cuts the flow from the study objectives and the methodology, as well as its problematization.

Methods

Line 154-162: this can just be seen in Figure 1.

-          Type-o’s in Figure 1. Fix! Also, the use of color in the figure is irrelevant.

Line 190: provide citation for Cohen’s interpretation

Table 2, age, others, intervention: 31.66 years? I think it would not hurt clarifying it.

Table 3, cognition, “ben-efits” why not just put it below instead of breaking it?

Line 242: when p values fall below 0.001, the = sign is redundant.

Figure 3 has a low quality. Also, p values do not equal 0.000, they represent probabilities. Therefore, they should be written as < 0.001. Ref boti N et al.(2019) should capitalize the author’s last name. Finally, there are symbols that are not explained in the figure’s legend.

Line 245-250: why not using the effect size’ symbol instead of “effect size”? Also, p of p values should be in italics.

Line 254: There should be a “Note:…” so the reader knows that belongs to the table.

Figure 4 has a low quality.

Discussion

Line 275: Showed*

Line 278: it is either publications of studies conducted in their country. Authors should be clear about this.

Line 280-282: The authors also may point out to the reader how politics on sexual education based on abstinence could play a role in these stats.

Line 287: studies do not “last”, their interventions did

Line 290-292: I believe this last sentence does not add much from what is previously stated.

Line 296: “are able to…” instead of “can”.

Line 307: Authors use STI and STD referring to the same phenomena. Whereas they are related, I point this to the authors to either pick one or explain why they use both.

Line 316: The citation provided is good. I just believe there should be, at least, more than one here.

Line 327: “greatest” is not precise.

Note: I fail to read anything on sexual debut, which is usually measured in sexual education programs or stats, often seen to be delayed when sexual education is given.

Line 364-368: I do not agree with this paragraph. I believe the systematic review and meta-analysis highlight other types of contributions. The ones mentioned pertain more to the studies reviewed.

Minor spelling mistakes or choose of tenses that I did not find appropriate. Yet, nothing the very same authors cannot handle.

Author Response

Dear reviewer:

We wish to re-submit the manuscript titled “Effects of Comprehensive Sexuality Education Programs on Children and Adolescents: A Systematic Review and Meta-Analysis” The manuscript ID is healthcare-2427938.

Thank you for your thoughtful review. I revised the manuscript based on your comments. Your advice was very helpful to advance my manuscript.

The responses to the reviewers’ revision requests have been prepared in a separate file, and the manuscript has been edited in response to these revision requests. Please see the attachment.

Thank you for your consideration.

Title

I believe “sexual” education is better suited.

The commonly used term for CSE is 'Comprehensive Sexuality Education', so I will maintain the term 'Sexuality'.

Abstract

Line 12: A study goal is not its rationale’s background. If background is required, then it should be along the lines of the context of the study.

I have revised the sentence to reflect the background of the study.

Introduction

Line 32: it may not hurt to add that sex education should be age-appropriate and culturally relevant.

I have incorporated the points mentioned in your comment.

Line 42: “the awareness of one’s own sexuality…” is better suited than “sexual consciousness”. Such sentence should have a reference. I recommend Pfaus et al., 2012 Figure 9 to see how this process is developed. 10.1007/s10508-012-9935-5

I have revised the content based on the points mentioned in your comment and added relevant references.

Line 58: the authors used in line 56 the term, they should use the acronym STI there.

I have added STI to line 56 and revised line 58 to STI in your comment.

Line 61: I think, given the topic of the study, that the authors should disclose the age of the women whom they speak in the citation.

I have added the age range of women, 'age 12 to 25' as mentioned in the citation.

Line 80-82: I believe the study has much more value than those—good reasons—mentioned here, especially those regarding the meta-analysis.

I have added “However, a broad analysis encompassing a wide range of countries and age groups is necessary to understand the effects of CSE in various cultural and social contexts.”.

Line 87-97: To identify…

I have consolidated the information about the objectives into a single objective, resulting in the deletion of the sentence.

Line 98: whereas what is described in this section is helpful and valuable, I think it should be embedded in the previous section, for it cuts the flow from the study objectives and the methodology, as well as its problematization.

I have moved the section in your comment.

Methods

Line 154-162: this can just be seen in Figure 1.

- Type-o’s in Figure 1. Fix! Also, the use of color in the figure is irrelevant.

Other reviewer also commented on the figure 1, and the reviewer provided feedback on the figure as well. We decided to remove the figure as they were deemed unnecessary and detracted from readability.

Line 190: provide citation for Cohen’s interpretation

I added citation.

Table 2, age, others, intervention: 31.66 years? I think it would not hurt clarifying it.

I modified it

Table 3, cognition, “benefits” why not just put it below instead of breaking it?

It has been presented in a single line.

Line 242: when p values fall below 0.001, the = sign is redundant.

I modified it.

Figure 3 has a low quality. Also, p values do not equal 0.000, they represent probabilities. Therefore, they should be written as < 0.001. Ref boti N et al.(2019) should capitalize the author’s last name. Finally, there are symbols that are not explained in the figure’s legend.

I have made corrections based on your feedback. Thank you for your thorough review.

Line 245-250: why not using the effect size’ symbol instead of “effect size”? Also, p of p values should be in italics.

I modified it.

Line 254: There should be a “Note:…” so the reader knows that belongs to the table.

I have added a note below the table.

Figure 4 has a low quality.

I have used the CMA program to generate high-quality images, which are expertly created originals. Due to the large number of papers involved, manually drawing them would likely result in lower quality.

Discussion

Line 275: Showed*

I modified it.

Line 278: it is either publications of studies conducted in their country. Authors should be clear about this.

Regarding your comment on Line 278, I understand your concern about the need for clarity. In response, I have revised the sentence to make it explicit that the statement refers to the United States.

Line 280-282(279-281): The authors also may point out to the reader how politics on sexual education based on abstinence could play a role in these stats.

Regarding your comment on Lines 280-282, I understand the importance of addressing how the political stance on sex education, particularly those advocating for abstinence-based education, could influence the statistics I have discussed. To address your comment, I have added sentences.

Line 287: studies do not “last”, their interventions did

I have revised the sentence to reflect your comment.

Line 290-292: I believe this last sentence does not add much from what is previously stated.

I have revised the sentence to reflect your comment.

Line 296: “are able to…” instead of “can”.

I modified it.

Line 307: Authors use STI and STD referring to the same phenomena. Whereas they are related, I point this to the authors to either pick one or explain why they use both.

I have revised from STD to STI. I pick and use only the term ‘STI’.

Line 316: The citation provided is good. I just believe there should be, at least, more than one here.

I have added additional citations in accordance with your comment.

Line 327: “greatest” is not precise.

I have revised from greatest to particularly notable.

Note: I fail to read anything on sexual debut, which is usually measured in sexual education programs or stats, often seen to be delayed when sexual education is given.

Thank you for your valuable comment on our manuscript. The precise numerical value for the age of sexual debut has been described differently across studies. Our research is a study that investigates the impact of sexual education through systematic review and meta-analysis. An important finding from our study is that sexual education can delay the age of first sexual intercourse. We have acknowledged the validity of the reviewer's comment in this regard. However, providing an exact numerical representation of the age of first sexual intercourse can compromise the readability of the research findings. Therefore, we have made a decision to describe the impact of sexual education on delaying the age of sexual debut without including specific numerical expressions. This decision was made through discussions among the researchers.

Line 364-368: I do not agree with this paragraph. I believe the systematic review and meta-analysis highlight other types of contributions. The ones mentioned pertain more to the studies reviewed.

I have revised the paragraph to reflect your comment.

Comments on the Quality of English Language

Minor spelling mistakes or choose of tenses that I did not find appropriate. Yet, nothing the very same authors cannot handle.

Thank you for your feedback on the quality of the English language in our manuscript. We apologize for any minor spelling mistakes or inappropriate choice of tenses that you may have come across. I have carefully proofread and made corrections for any typos or grammatical errors. If you find any parts that are still difficult to read or need further improvement, please let me know, and I will make the necessary revisions.

Reviewer 2 Report

REVIEW 1: Effects of Comprehensive Sexuality Education Programs on Children and Adolescents: A Systematic Review and Meta-Analysis

The authors have conducted a review aimed at understanding aspects related to sex education. This topic is interesting and very necessary nowadays as the trend of risky sexual behaviors is on the rise.

This manuscript is interesting although not enough information is provided to reproduce it and it is not up to date.

Other deficiencies detected:

INTRODUCTION

The introduction should not be to have subsections.

Authors should include some epidemiological data to support their claims. For example, it would be good to include data on unwanted pregnancies.

There is redundant information in the objectives. It is also unclear whether there is one general objective and several specific ones. The authors should review this topic. Also, the objectives are always the last thing in the "Introduction

Figures should not be included in the “introduction”.

MATERIALS AND METHODS

Search Strategy:

The equation is poorly stated. When the equation is entered literally as written in the manuscript, the number of references in each database does not appear. The authors should revise this.

Revisions made more than 6 months ago must be updated. This manuscript has analyzed until September 30, 2022.

Selection and exclusion criteria for literature:

The inclusion criteria are poorly defined. The study cannot be reproduced with the information provided by the authors.

RESULTS

The authors have not provided a narrative description of the "results". Therefore, there is no systematic review, only a meta-analysis.

Authors cannot use references to "results" in the "introduction" or "discussion", this is wrong.

Authors should do a reverse search to get more results.

DISCUSSION

The authors have not compared their "results" with other references. The authors should look for references to justify their "results".

REFERENCES

Many bibliographies are obsolete and some citations are incomplete. The bibliographic citations used are more than 5 years old (48% not including "results"). Authors should update the "introduction" and "discussion" references.

The authors provide few bibliographical references. In a systematic review with meta-analysis this is not logical. Nor are the references of the "results" provided.

Author Response

Dear reviewer:

We wish to re-submit the manuscript titled “Effects of Comprehensive Sexuality Education Programs on Children and Adolescents: A Systematic Review and Meta-Analysis” The manuscript ID is healthcare-2427938.

Thank you for your thoughtful review. I revised the manuscript based on your comments. Your advice was very helpful to advance my manuscript.

The responses to the reviewers’ revision requests have been prepared in a separate file, and the manuscript has been edited in response to these revision requests. Please see the attachment.

Thank you for your consideration.

INTRODUCTION

The introduction should not be to have subsections.

→ I removed the subsection based on your comment.

Authors should include some epidemiological data to support their claims. For example, it would be good to include data on unwanted pregnancies.

→ I added epidemiological data about unwanted pregnancies and its reference.

There is redundant information in the objectives. It is also unclear whether there is one general objective and several specific ones. The authors should review this topic. Also, the objectives are always the last thing in the "Introduction

→ I have consolidated redundant information about the objectives into a one general objective based on your comment. Additionally, I have moved the objectives to the end of the 'Introduction' section.

Figures should not be included in the “introduction”.

→ I removed Figure1 based on your comment.

MATERIALS AND METHODS

Search Strategy:

The equation is poorly stated. When the equation is entered literally as written in the manuscript, the number of references in each database does not appear. The authors should revise this.

→ I modified it.

Revisions made more than 6 months ago must be updated. This manuscript has analyzed until September 30, 2022.

→ We conducted the research in accordance with the study plan, obtaining approval from the IRB (Institutional Review Board) of the researcher's affiliated school. Seven authors synchronized their time and gathered in the laboratory for this study, carrying out cross-validation and progressing with the research as planned. If we were to search for additional papers after October 2022 and repeat this process, it is expected to consume a significant amount of time once again. Your suggestion is appreciated, and it would be beneficial to supplement through further research in the future. Thank you for your valuable input.

The inclusion criteria are poorly defined. The study cannot be reproduced with the information provided by the authors.

→ We refined the inclusion criteria to provide clearer guidelines.

RESULTS

The authors have not provided a narrative description of the "results". Therefore, there is no systematic review, only a meta-analysis.

Authors cannot use references to "results" in the "introduction" or "discussion", this is wrong.

Authors should do a reverse search to get more results.

→ Thank you for your comment regarding the addition of the results section to our manuscript. We appreciate your suggestion and are pleased to inform you that we have carefully reviewed your feedback and made the necessary revisions. The updated version of the manuscript now includes a comprehensive results section, providing a detailed description of our findings (Table S1). If you have any further comments or suggestions, please do not hesitate to let us know. We highly value your expertise and look forward to your continued guidance in enhancing the quality of our research.

DISCUSSION

The authors have not compared their "results" with other references. The authors should look for references to justify their "results".

→ We appreciate your comment and are pleased to inform you that we have carefully reviewed your feedback and made the necessary revisions. The updated version of the manuscript now includes a comprehensive results section, providing a detailed description of our findings (Table S1).

REFERENCES

Many bibliographies are obsolete and some citations are incomplete. The bibliographic citations used are more than 5 years old (48% not including "results"). Authors should update the "introduction" and "discussion" references.

The authors provide few bibliographical references. In a systematic review with meta-analysis this is not logical. Nor are the references of the "results" provided.

→ We appreciate your comment regarding the outdated bibliographies and incomplete citations in our manuscript.  In response to your feedback, we have conducted a thorough review of our references and made necessary revisions to ensure that they are up-to-date and relevant. We have updated the "introduction" and "discussion" sections with recent and pertinent citations, taking into account the importance of including current literature in a systematic review with meta-analysis. Furthermore, we have addressed the issue of missing references in the "results" section, ensuring that appropriate citations are now provided to support our findings.

Round 2

Reviewer 2 Report

REVIEW 2: Effects of Comprehensive Sexuality Education Programs on Children and Adolescents: A Systematic Review and Meta-Analysis

Dear Editor, the authors have improved the article but there are still deficiencies that they have not fixed. In my opinion they cannot be fixed without having to rewrite the article again from scratch.

Deficiencies that remain:

INTRODUCTION

The paragraph that talks about objectives is still confusing. First some objectives are mentioned and in the last sentence it says that they are others. This is inconsistent.

AGAIN: AUTHORS CANNOT USE REFERENCES TO "RESULTS" IN THE "INTRODUCTION" OR "DISCUSSION", THIS IS WRONG.

MATERIALS AND METHODS

Search Strategy:

AGAIN: Revisions made more than 6 months ago must be updated. This manuscript has analyzed until September 30, 2022.

Selection and exclusion criteria for literature:

AGAIN: The inclusion criteria are poorly defined. The authors do not say since when they include the studies or in what language, for example. The study cannot be reproduced.

RESULTS

AGAIN: The authors have not provided a narrative description of the "results". Therefore, there is no systematic review, only a meta-analysis.

AGAIN: AUTHORS CANNOT USE REFERENCES TO "RESULTS" IN THE "INTRODUCTION" OR "DISCUSSION", THIS IS WRONG.

DISCUSSION

AGAIN: AUTHORS CANNOT USE REFERENCES TO "RESULTS" IN THE "INTRODUCTION" OR "DISCUSSION", THIS IS WRONG.

REFERENCES

AGAIN: Authors should update the "introduction" and "discussion" references.

Author Response

Dear Reviewer,

We would like to resubmit the manuscript titled "Effects of Comprehensive Sexuality Education Programs on Children and Adolescents: A Systematic Review and Meta-Analysis" with revised title and type. The manuscript ID is healthcare-2427938.

We have revised the title to "Meta-Analysis of Effects on Comprehensive Sexuality Education Programs on Children and Adolescents" instead of "Effects of Comprehensive Sexuality Education Programs on Children and Adolescents: A Systematic Review and Meta-Analysis." Additionally, we would like to change the type of the manuscript from "Systematic Review" to "Review."

We have made revisions to the manuscript based on your feedback. Furthermore, we have written the report with a primary emphasis on the results obtained from the meta-analysis.

Your advice has been extremely helpful in improving the quality of our paper. We appreciate your thoughtful review.

Thank you for your consideration.

INTRODUCTION

The paragraph that talks about objectives is still confusing. First some objectives are mentioned and in the last sentence it says that they are others. This is inconsistent.

AGAIN: AUTHORS CANNOT USE REFERENCES TO "RESULTS" IN THE "INTRODUCTION" OR "DISCUSSION", THIS IS WRONG.

Thank you for your valuable feedback. We greatly appreciate it and have carefully considered it during the revision process.

We have restructured the content to address the issue of inconsistency and ensure a logical flow. We emphasized the importance of sexual education for children and adolescents and the development of holistic growth and healthy self-concepts. Additionally, we included specific information and statistical data on the challenges and issues related to sexual problems among children and adolescents worldwide. Furthermore, we introduced the concept of comprehensive sexuality education (CSE) as a curriculum that covers cognitive, emotional, physical, and social aspects of sexuality.

In summary, we have thoroughly revised the introduction section to address concerns regarding clarity and consistency.

Additionally, we have also removed the content related to the "Results" mentioned in the description.

We believe that the revised introduction significantly enhances the quality of our paper, and we hope it meets your expectations. Once again, we express our gratitude for your valuable feedback.

MATERIALS AND METHODS

Search Strategy:

AGAIN: Revisions made more than 6 months ago must be updated. This manuscript has analyzed until September 30, 2022.

We have updated the content from September 30, 2022, to June 14, 2023. One additional study was included in the analysis, and the updated information has been added to the "Literature search and selection" section. We have also made revisions to the PRISMA flow chart (Figure 1).

Selection and exclusion criteria for literature:

AGAIN: The inclusion criteria are poorly defined. The authors do not say since when they include the studies or in what language, for example. The study cannot be reproduced.

To provide clearer guidelines, we have improved the inclusion criteria. Specifically, we have added a method for categorizing the components of CSE.

RESULTS

AGAIN: The authors have not provided a narrative description of the "results". Therefore, there is no systematic review, only a meta-analysis.

AGAIN: AUTHORS CANNOT USE REFERENCES TO "RESULTS" IN THE "INTRODUCTION" OR "DISCUSSION", THIS IS WRONG.

We decided to focus on a meta-analysis that excluded systematic literature reviews from the existing systematic literature reviews and focused solely on meta-analysis.

We have revised the title to "Meta-Analysis of Effects on Comprehensive Sexuality Education Programs on Children and Adolescents" instead of "Effects of Comprehensive Sexuality Education Programs on Children and Adolescents: A Systematic Review and Meta-Analysis." Furthermore, when writing the report, we have placed a primary emphasis on the results obtained from the meta-analysis.

We have made modifications to the numerical values in the tables and figures, taking into account the analysis results that include the additional studies. Additionally, we have addressed the feedback regarding the low quality of the forest plot by generating a new figure. Furthermore, to facilitate easy visual assessment of the effect sizes for Variables and Components, we have included a forest plot of the effect sizes as well.

DISCUSSION

다시: 저자는 "서론" 또는 "토론"에서 "결과"에 대한 언급을 사용할 수 없습니다. 이것은 잘못된 것입니다. 

"서론" 또는 "토론" 섹션에서 "결과"에 대한 언급 사용에 관한 문제를 제기해 주셔서 감사합니다. 실수에 대해 사과드리며 이 문제를 해결하기 위해 필요한 수정을 했습니다.

또한 개정된 논의 부분에는 새로 추가된 연구를 활용한 분석과 관련된 내용을 포함시켰다.

개정의 필요성을 지적해 주신 검토자에게 감사드리며, 업데이트된 토론 섹션이 이제 연구 결과를 정확하게 반영한다고 믿습니다.

참조

다시: 저자는 "소개" 및 "토론" 참조를 업데이트해야 합니다.

귀하의 피드백에 대한 응답으로 참조를 철저히 검토하고 최신 정보와 관련성을 보장하기 위해 필요한 수정 작업을 수행했습니다.